

# A simple guide from machine learning outputs to statistical criteria in particle physics

**Charanjit Kaur Khosa[1⋆], Veronica Sanz[2,3†] and Michael Soughton[3‡]**

**1** H.H. Wills Physics Laboratory, University of Bristol, Tyndall Avenue, Bristol BS8 1TL, UK
**2** Departament de Física Teòrica and IFIC, Universitat de València-CSIC,
E-46100, Burjassot, Spain
**3** Department of Physics and Astronomy, University of Sussex, Brighton BN1 9QH, UK

⋆ Charanjit.Kaur@bristol.ac.uk , † V.Sanz@sussex.ac.uk , ‡ M.Soughton@sussex.ac.uk

## Abstract

In this paper we propose ways to incorporate Machine Learning training outputs into a study of statistical significance. We describe these methods in supervised classification tasks using a CNN and a DNN output, and unsupervised learning based on a VAE. As use cases, we consider two physical situations where Machine Learning are often used: high-$p_T$ hadronic activity, and boosted Higgs in association with a massive vector boson.

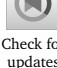

# 1 Introduction

The world of Particle Physics is a world of discoveries or, in their absence, of limit setting. We perform measurements in the hope to establish firm evidence of the existence of a process. To do so, we quote a level of statistical significance with some agreed convention e.g., a 3-$\sigma$ confidence level would mean a hint, and a discovery is only reached at 5-$\sigma$. In the context of Particle Physics, these criteria help us validate the Standard Model (SM) paradigm, or find new phenomena.

To make statements about discovery, we rely on very strict statistical criteria [1], i.e. those $\sigma$'s are obtained after a lot of thinking on possible sources of systematic experimental errors and theoretical uncertainties. At the Large Hadron Collider (LHC), the experimental collaborations have set an even higher bar on claims for discovery, including additional levels of strictness like data-driven estimations, e.g. [2], and the *look elsewhere effect* [3].

In this context, Machine Learning (ML) is erupting as an exciting tool for experimental analyses and theoretical studies in Physics [4] and particularly in Particle Physics.[1] In its simplest form, ML is used as supervised learning to perform classification. But the quick development of ML techniques elsewhere is driving new uses of ML in Particle Physics, beyond classification or regression. For example, the potential of unsupervised methods, specifically Deep Generative Modeling, is being explored in many contexts: from Anomaly detection [5] using Variational AutoEncoders (VAE), to Monte Carlo Generation where the concept of data amplification is applied to GANplify [6–8] samples. Machine Learning is also thoroughly used in achieving effective inference [9, 10] and approximating likelihoods [11–13], uncertainty quantification [14, 15] and now even tackling symmetry identification [16–19].

This increase of ML analyses brings home the question of how to link ML outputs, typically based on training with large amounts of synthetic data, with the traditional criteria for statistical significance. The answer to this question is not unique, as ML techniques are diverse and could be applied at many different levels in an analysis. In this paper we go some way to fill this gap, proposing simple ways of translating ML outputs into statistical criteria. We hope this first approximation leads to the development of better, more sophisticated methods, which fully use the power of expression of Deep Learning modelling.

The paper is organised as follows. In Sec. 2 we discuss how to translate the typical classification problem in ML into a statistical significance based on the *Cousins et al.* proposal [20, 21] of computing the Log-Likelihood Ratio (LLR). We will apply this procedure to Hypothesis Testing. In Sec. 3 we will propose a method to use the outputs of unsupervised ML models, specif-

---

[1]To gauge the extent of ML in this area, have a look at the living review in https://github.com/iml-wg/HEPML-LivingReview.

ically VAEs and GANs, to estimate a separation significance. We will present our conclusions in Sec. 4.

The numerical computations of hypothesis testing presented in this paper can be found in the following `GitHub` repository: https://github.com/high-energy-physics-ml/hypothesis-testing.

# 2 Supervised Machine Learning and hypothesis testing

A common task within Machine Learning is that of classification. This requires a labelled dataset to train on and it will typically output a predicted probability $P$ of some input data, which ideally would closely match the label for testing data. From this learning, metrics such as accuracy, F-score, and ROC curves can be constructed [22]. For an event-by-event analysis, such quantities are useful. However, if we want to claim discovery of new phenomena, then we will want to perform a proper hypothesis test. Here we shall outline how one can perform a simple hypothesis test using the predicted probability from a classification task.

## 2.1 Simple hypothesis testing

With a given set of data, we often want to ask a simple questions with a 'yes' or 'no' answer. For example, 'does this data contain new physics events?' is a question with a binary answer, testing a hypothesis $H_0$ ('There is no New Physics') with the alternative hypothesis $H_1$ ('There is New Physics').

To test the hypotheses we require some test statistic which measures how well data favours either hypothesis. If one were to obtain data which yields some value for the test statistic, we can quantify how much such a value agrees with the test statistic values expected under either hypothesis. We refer the reader to Refs. [1, 23–25] for very pedagogical resources on hypothesis testing in the context of Physics, and here we will just briefly summarize the framework and set the notation.

We can set a cutoff value for this test statistic that defines the point at which the null is accepted or rejected. In doing so we will have, by construction, a probability $\alpha$ that if the null *were* true of incorrectly rejecting it (also called a type I error, or false positive) which is the *significance level* of the test. We can also obtain a probability $\beta$ if the alternative *were* true of incorrectly not rejecting the null (not accepting the alternative) also called a type II error, or false negative. The *power* of the test is defined as $1 - \beta$. Additional background details on these two types of errors are provided in Appendix B.

The Neyman-Pearson lemma [26, 27] states that for a test between two simple hypotheses the optimal test statistic is the Likelihood Ratio. As is typical in Particle Physics, we can perform simulations to generate synthetic data under the assumption that $H_0$ or $H_1$ are correct. Then, one can obtain a Likelihood Ratio for both types of data (or in other words the Likelihood *given* either hypothesis being true).

More generally, if we have a Likelihood that depends on some parameter $\boldsymbol{\theta}_i$ (which can be a a vector or a single value) given some hypothesis $H_i$ with $i = \{0, 1\}$ *is* true, then the Likelihood Ratio is

$$\lambda_{H_i} = \frac{L(\boldsymbol{\theta}_0 \mid H_i)}{L(\boldsymbol{\theta}_1 \mid H_i)}. \tag{1}$$

Note that the *given $H_i$* notation here is synonymous with saying *given observed data that happened to be generated under $H_i$* which is sometimes written $L(\boldsymbol{\theta} \mid \mathcal{D})$ or $L(\boldsymbol{\theta} \mid x)$ in other literature, but when we take the Likelihood Ratio it is necessary to write the hypotheses explicitly.

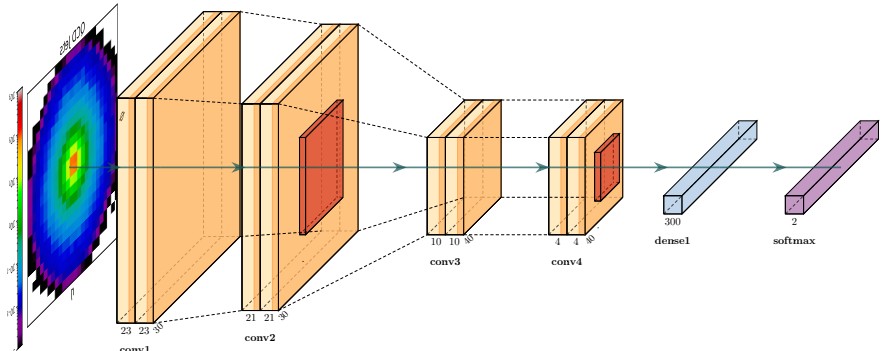

Figure 1: The CNN used. A jet image of dimensions 25x25 is fed through two 2D Convolutional layers and a 2D Max Pooling layer followed by two more two 2D Convolutional layers and another 2D Max Pooling layer, each layer reducing the size of each dimension. Finally this is flattened and fed into a Densely connected layer before reaching the output layer with a shape of 2. The output layer use Softmax activation and all other layers use ReLU.

We will actually work with the Log-Likelihood Ratio (LLR)

$$\Lambda_{H_i} = -2\ln\lambda_{H_i}. \tag{2}$$

We obtain our Likelihoods from the Probability Density Functions (PDFs) that describe the distribution of data, which are comprised of two terms: a term for the Poisson PDF, since the number of events follows a Poisson distribution, and a term for the PDF $p$ of whatever observed quantity is used.[2]

This Likelihood is called a marked Poisson Likelihood [24]. This total PDF can describe any general detector experiment and the observed quantity could be e.g. $\eta$, $\phi$, $p_T$ etc., however we shall here say the observed quantity of the PDF in the second term is the predicted output from the binary classification Machine Learning task, a probability $P$.

## 2.2 Use case I: Top vs QCD Jet images

To explain the procedure to incorporate ML outputs into the significance computation, we start with a benchmark for top tagging studies. This is a well-known case of classification task in the LHC, where high-energy hadronic tops are tagged against the main competing background of QCD. The use of ML, and in particular Convolutional Neural Networks (CNNs), to improve top tagging was proposed in Ref. [28] and has been thoroughly studied, e.g. [29, 30]. This task is usually presented as a image to label task, where the labels are Top and QCD classes.

We use a Convolutional Neural Network (CNN) which, once trained, will attempt to classify any given jet image as either QCD or Top. Details on how we generated the events and produced the images can be found in Appendix A. We use a combination of convolutional layers and pooling layers as well as a dense layer, and finally an output using a softmax activation as depicted in Figure 1. All other activations are ReLU. We train on 40K each of QCD and Top

---

[2]Note that in statistics literature PDFs are often called *models* - here we shall refer to them as such and distinguish Particle Physics models and Machine Learning models.

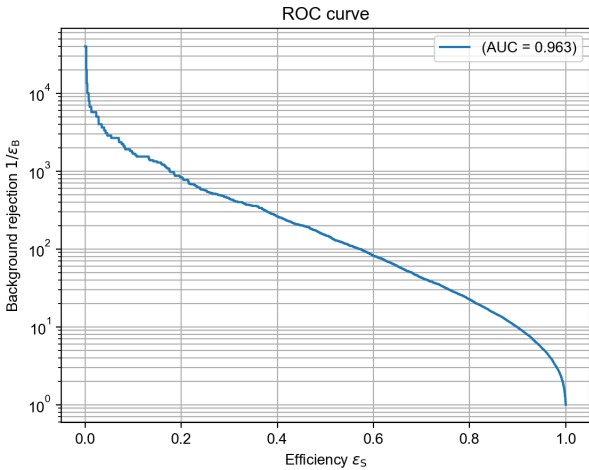

Figure 2: ROC curve for the Top vs QCD task.

jet images. The CNN architecture was optimised by considering accuracy and ROC curves from testing data of 20K of each.

With the CNN trained, we can feed it new jet images and it will return a value $P$ of the image belonging to either class. Since this $P$ is given in the one-hot encoding form, we translate it into a predicted probability $P(\text{Top})$ of the image being a Top jet.

We perform Bootstrapping [31] to evaluate the change in model performance over different samplings of the training and test data. Bootstrap Aggregating, also called Bagging, involves sampling from the dataset many times with replacement (i.e. performing the random train-test split and running the model many times) to obtain a distribution of the model performance. Doing so allows us to obtain a statistical measure of how well the model is performing, since in practice we can expect a Machine Learning model's performance to fluctuate slightly. Over 1000 Bootstraps we find that our CNN performs with an average accuracy of 87.9% with a 95% confidence interval of 87.1% and 88.5%.

After the training is done, the algorithm takes an image and outputs a probability of belonging to the QCD or Top classes. One can represent thee learning of the algorithm using a classical ROC curve which show the different working points for true and false positive rates as we vary the probability threshold.

But in Particle Physics it is often more convenient to plot a modified version of the ROC curve with background rejection and signal acceptance relations, as shown in Fig. 2. One could use this representation of the output of the ML task to understand which optimal cuts maximize interesting quantities, like the ratio of signal vs background events, or some other measure of approximate significance. This procedure would be naive, especially with low statistics, but a good first approximation to understand the gain in using ML.

In this paper we want go further and provide a more sophisticated, yet approachable, way to use ML outputs. In the next three sections, we will start by identifying ML outputs as probability distribution functions, use them to build a robust test statistic method and translate into significance.

### 2.2.1 Obtaining PDF distributions for QCD and Top data

To get the LLR distributions we must first obtain the PDF $p_{H_j}$ under the different hypotheses $j = $ QCD and QCD+Top.

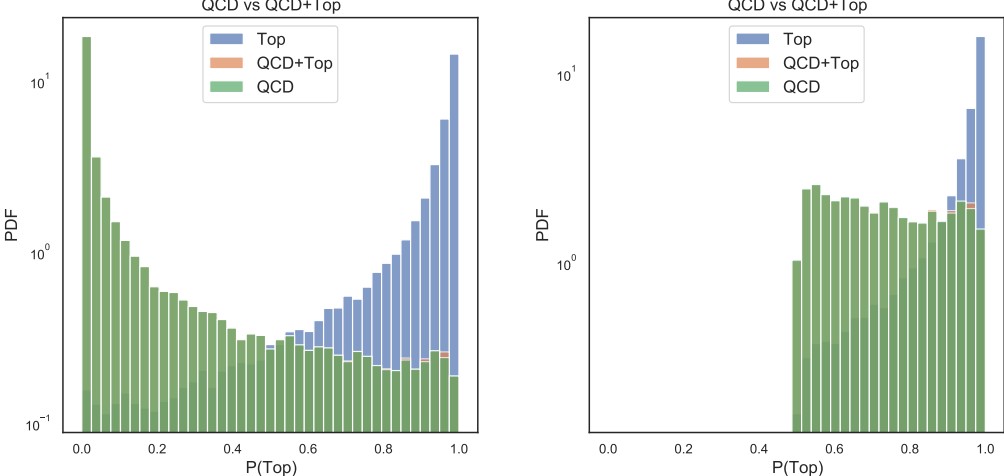

Figure 3: *Left plot:* PDF distribution of the output of the classification task, P(top), for samples with pure top events, pure QCD events and a mixture of QCD and top events weighted by their respective cross-sections. *Right plot:* same as left plot but with $P_{cut} > 0.5$.

We will adopt the distribution of the ML classifier output as the PDF distribution of the data. To build the PDF, we train the algorithm with QCD and Top images. After freezing the trainig, we feed a large number of new jet images into the CNN algorithm to provide a probability of an image to be of a class. In our case, an image would lead to an output $P(\text{Top}) = 1 - P(\text{QCD})$. We build this distributions for a dataset containing only QCD images, only Top images, and for a dataset with QCD images mixed with some Top images with the ratio of QCD to Top images set by their cross-sections, as shown in the left panel in Figure 3.

In the figure we show the pure QCD and Top PDFs as well as the mixed PDF. The pure PDFs show what one would expect from a good performing classifier with most QCD events having $P(\text{Top})$ around zero and most Top events having $P(\text{Top})$ around one. Notice that the mixture of Top events with the QCD events only slightly changes the PDF from the pure QCD one. This is because the Top cross-section is much smaller than the QCD one. The LLR will essentially be evaluating this difference between the QCD and QCD + Top PDFs, however we shall see that is still able to do this to a good sensitivity.

Given the overwhelming dominance of the QCD events on an unbiased sample, one could think alternative PDF inputs for the LLR computation which could more efficiently capture the desired signal. One possibility would be to consider the output distribution with some cut on the probability. For example, one could focus on the distribution of events whose classifier $P(\text{Top})$ is above some threshold, as shown in the right panel of Fig. 3. In the next section we will show the effect of this possible modification in the input PDF in the overall LLR.

The distributions shown in Fig. 3 correspond to the average over many runs for which we have done bootstrapping to account for ML fluctuations. In Fig. 4 we show the effect of this procedure in the $P(\text{Top})$ output.

### 2.2.2 Obtaining LLR distributions for QCD and Top data

After building the PDFs, we can now turn our attention to building the LLR. As mentioned before, we will consider the marked Poisson Likelihood [24], which consists on two terms, a term for the Poisson distribution of $n$ events, and a term which accounts for the PDF density,

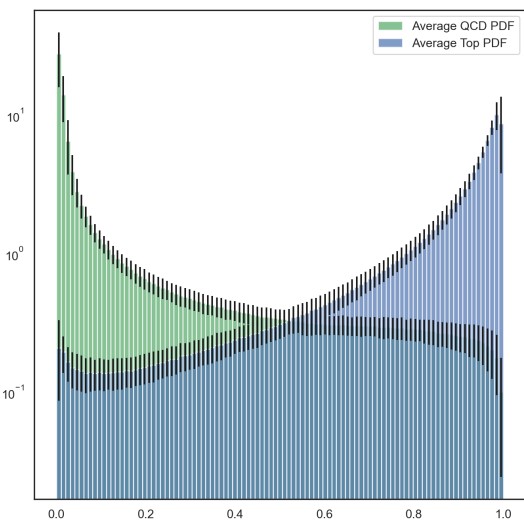

Figure 4: The effect on $P(\text{Top})$, the Machine Learning output chosen as PDF, of boostrapping. Solid distributions represent the average and the black lines represent the range of uncertainty in the average.

which in our case is given by ML outputs. The likelihood can be expressed as follows,

$$L(b, s_j, \boldsymbol{P}_j(\text{Top}) \,|\, H_i) = \frac{(b + s_j)^n \exp(-(b + s_j))}{n!} \cdot \prod_{k=1}^{n} p_{H_j}\left(P_k(\text{Top}) \,|\, H_i\right). \tag{3}$$

Let us discuss both terms. The Poisson term is the Poisson PDF values where data generated under $H_i$ is observed ($H_i$ *is* true) but we assume $H_j$ to be true (hence the parameters $b$ and $s_j$). This can be written generally as $\text{Pois}\left((n \,|\, H_i) \,|\, H_j\right) \equiv \text{Pois}_{H_j}(n \,|\, H_i)$, where the $H_j$ subscript is a shorthand notation to denote 'the PDF *assuming* $H_j$ is true'. Similarly the ML term is the ML output PDF values where data generated under $H_i$ is observed (the data corresponding to the $P_k(\text{Top})$ observed) but $H_j$ is assumed to be true, and the same shorthand notation is used.

The Poisson term depends on $n$, the number of observed events for a given experiment, such that under the null ($H_0 =$ 'QCD only') hypothesis $\mathbb{E}[n] = b$, and under the alternative ($H_1 =$ 'QCD + Top hypothesis'), $\mathbb{E}[n] = b + s$. Here $b$ represents the mean number of background events, and $s$ the mean number of signal events. Generally we can write the number of signal events as $s_j$ where $s_0 = 0$ and $s_1 = s$ to make compact our notation.

The second, Machine Learning output, term in Eq. 3 depends on the ML outputs after training over a large number of simulated events. The resulting distributions for the output $P(\text{Top})$ are shown in Fig. 3. We will use them to obtain for each event $k$ ($k = 1 \ldots n$) a value of $P_k(\text{Top})$.

The total Likelihood is then written as the likelihood of obtaining a distribution with parameters $b$, $s_j$ and $\boldsymbol{P}_j(\text{Top})$, given data generated under $H_i$. Here $j = \{0, 1\}$ is an index for the Hypothesis $H_j$ which is independent of $H_i$.

For example, assuming $H_{j=0}$ was true, the events we would observe would be QCD only, and one would be sampling over these $n$ QCD events drawn from the green distribution. We would obtain a ML contribution to $L(b, s_j, \boldsymbol{P}_j(\text{Top}) \,|\, H_0)$ given by $\prod_{k=1}^{n} p\left(P_k(\text{Top}) \,|\, H_0\right)$. If the truth was $H_{j=1} = \text{QCD} + \text{Top}$, the procedure would be the same, but now the events would be drawn from the green distribution.

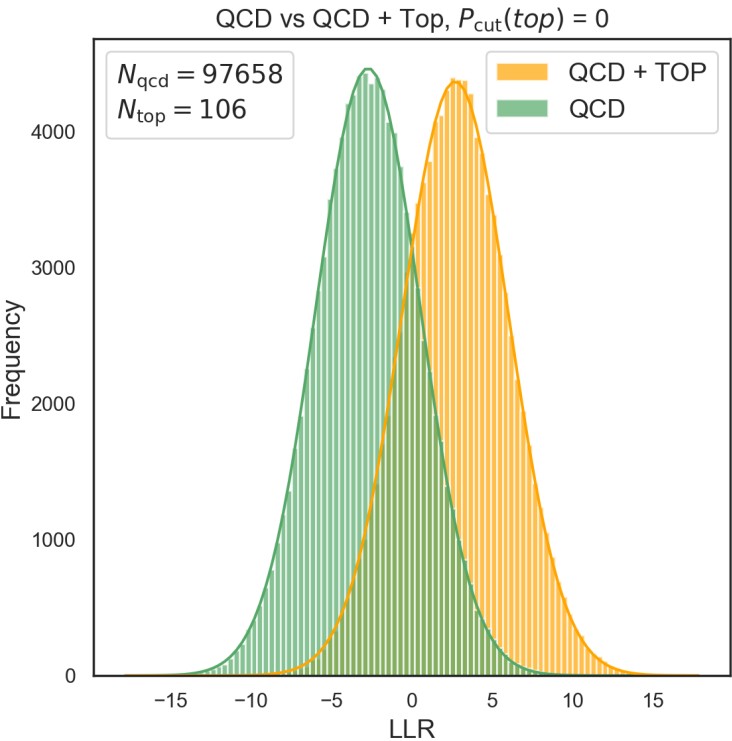

Figure 5: The Log-Likelihood Ratio for QCD and QCD+top at a integrated luminosity of 2 fb$^{-2}$ using 100k toy experiments.

Now, for this Likelihood we can write the Likelihood Ratios as

$$\lambda_{H_0} = \frac{L(b, s_0 = 0, \boldsymbol{P}_0(\mathrm{Top}) \mid H_0)}{L(b, s_1 = s, \boldsymbol{P}_1(\mathrm{Top}) \mid H_0)}, \tag{4}$$

and

$$\lambda_{H_1} = \frac{L(b, s_0 = 0, \boldsymbol{P}_0(\mathrm{Top}) \mid H_1)}{L(b, s_1 = s, \boldsymbol{P}_1(\mathrm{Top}) \mid H_1)}, \tag{5}$$

from which we will obtain the LLRs using equation (2). Notice that these Likelihoods can change under repeats of the experiment due to $n$ and $P_k(\mathrm{Top})$ subject to statistical variations.

We expect then that, if we were to repeat the same experiment a number of times, we would obtain some distributions for the Likelihood Ratios or Log-Likelihood Ratios. Indeed, with these PDFs we can obtain the Likelihood Ratios corresponding to equations (4) and (5) by sampling $n$ i.i.d. $p_{H_j}$ values (corresponding to different $P_k(\mathrm{Top})$) from them, with $n$ drawn from the Poisson distribution with means $b$ and $b + s$ respectively. Note that $b$ and $s$ are found from the cross-sections under different integrated luminosities after considering showering effects. We can build distributions of the LLRs by simulating a number of toy Monte Carlo experiments. We show the resultant distributions $f_0(\Lambda) \equiv f(\Lambda_{H_0})$ and $f_1(\Lambda) \equiv f(\Lambda_{H_1})$ in Figure 5 for 100k toy experiments, in this case with 97658 QCD events and 106 Top events (97764 QCD + Top combined) which corresponds to an integrated luminosity of 2 fb$^{-2}$. Note the good convergence to Gaussian distributions, which we fit to the histograms as expected from the central limit theorem.[3]

---

[3]Note that we plot the histograms of LLR values rather than their normalised PDFs in line with other physically motivated papers such aa [21, 32].

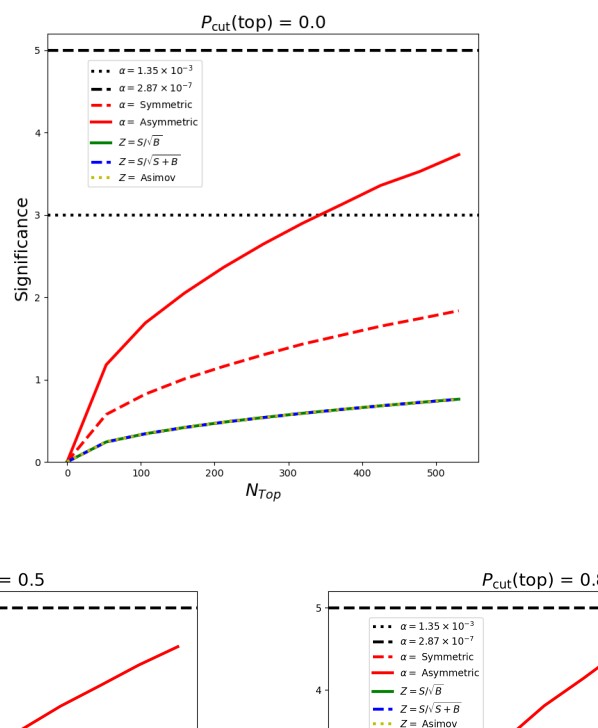

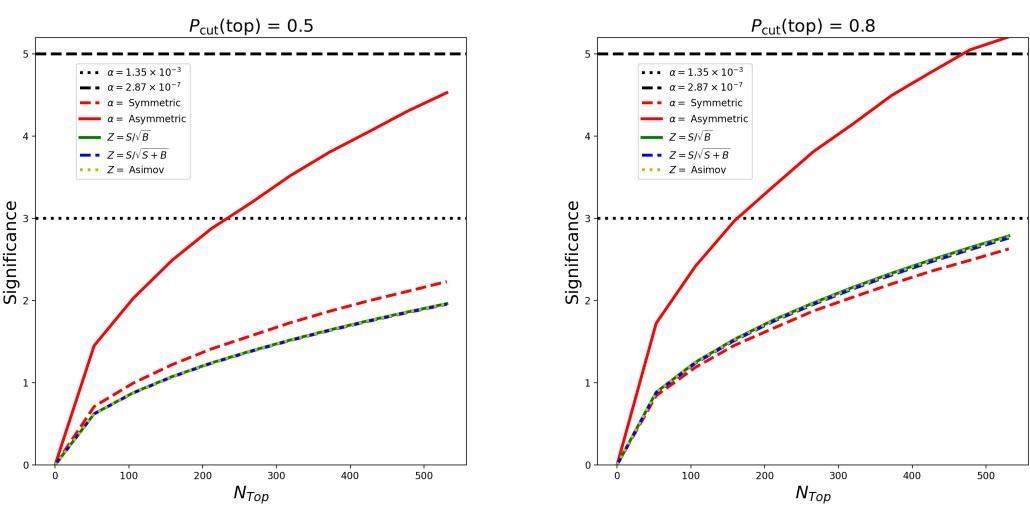

Figure 6: The significance level of symmetric and asymmetric tests for jet images classifier, expressed in terms of number of standard deviations of the Gaussian for $P_{\text{cut}}(\text{Top})=0$, 0.5 and 0.8, respectively. Standard discovery significances are also shown for comparison.

### 2.2.3 Obtaining a significance

The two distributions become more separated as the the number of events/luminosity increases, leading to a more confident statement in accepting or rejecting the null. From these distributions we can obtain the cutoff value which defines $\alpha$ and $\beta$. This cutoff value can be defined in various ways, leading to different conclusions for discovery which we explain in Appendix B. For the Top vs QCD classification, the results can be seen in Fig. 6.

This figure shows Significance as a function of the number of Top events present in the dataset. The number of Top versus QCD events is set by the relative cross-sections, see Sec. A. As expected, significance grows with the number of Tops present in the data, but this translates into a different value for significance depending on the criteria applied. In solid-red we show the results from an asymmetric criteria, whereas in dashed-red the result using a symmetric criteria. In this figure we also show the naive measures for significance based on various proxies for $Z = \text{Asimov}$, $s/\sqrt{b}$, and $s/\sqrt{s+b}$, which are all aligned in this background-dominated

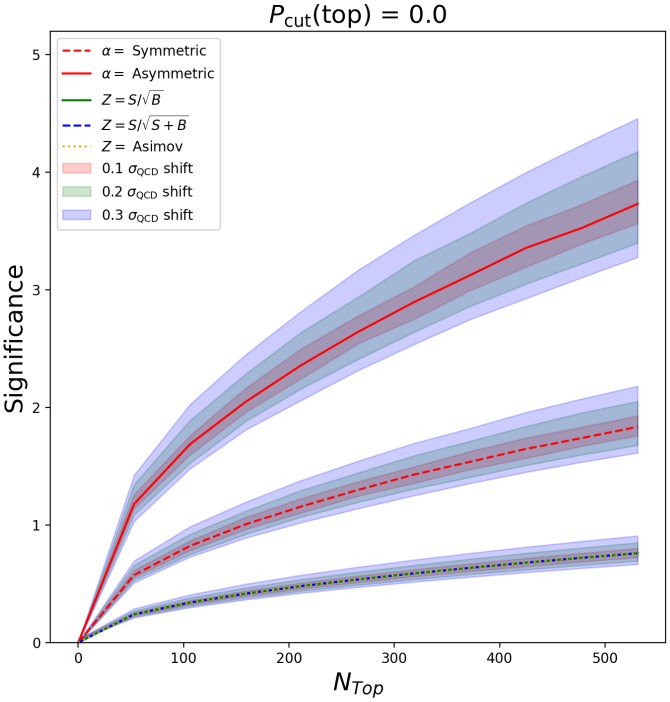

Figure 7: The significance levels as shown previously but now with errors due to uncertainty in the QCD cross-section, shown with 10%, 20% and 30% differences in cross-section.

case.

We also present the effect of applying a cut on the output classifier. From top to bottom, and left to right, we show results using the full output distribution, $P_{\text{cut}}(\text{Top})=0$ (top panel), and the effect of cutting on it, $P_{\text{cut}}(\text{Top})= 0.5$ and $0.8$ (bottom panel), respectively. Under all the significance criteria, overall significance improves as one cuts on the classifier. This improvement is limited, thought, as a very strong $P_{\text{cut}}(\text{Top})$ would significantly reduce the statistics.

### 2.2.4 The effect of uncertainties

Here we shall show the effects of different sources of uncertainties on the significance level. These uncertainties are not intended as a detailed assessment of sources of errors, but to show how the significance can change due to various effects.

An obvious source of uncertainty is from not having a precise measurement of the number of background events. So far we have been treating this as well known (the QCD background *is* well known to a good precision) however if it were not then it could affect the Likelihood Ratios. An estimate of uncertainty associated to this final state can be found in the ATLAS measurement [33], where one founds that for $p_T$(leading jet)$> 750$ GeV, the estimated uncertainties[4] range from 15 to 20% for statistical uncertanties and around 30% for systematics.

To give a sense of the effect of these uncertainties in the significance calculation, we can compute the LLR with different numbers of background events. In Figure 7 we show the

---

[4]See also the HEPDATA link to this study https://www.hepdata.net/record/ins1646686.

separation significance one would obtain had QCD cross-sections of ± 10%, 20% and 30% the given value been used (shown also for the Asimov significance). An alternative method to account for uncertainty in the number of background events would be to take it as a nuisance parameter for example as done in [25,34].

We also consider uncertainties which may arise from additional noise in the detector by smearing the jet images with a Gaussian filter[5] with a single parameter, a standard deviation $\sigma$, which it creates noise in surrounding pixels. As we increase $\sigma$ the image becomes more blurry, as shown in Figure 8. Nevertheless, we found that the CNN classification is robust

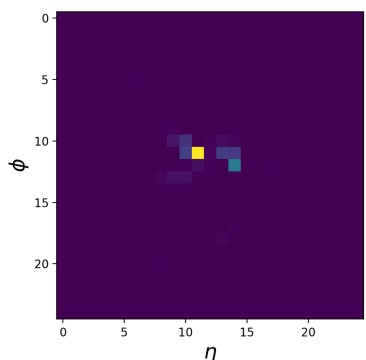 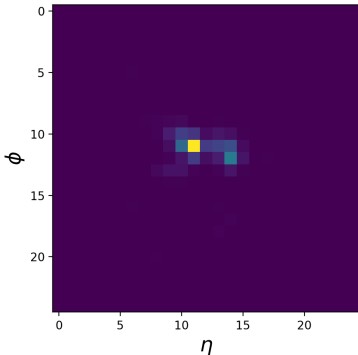

Figure 8: Example of a QCD image (left) and the same image after blurring with $\sigma$=0.5 (right).

under blurring. Indeed, despite varying $\sigma$ in a reasonable range up to around 0.5, we found no substantial change as compared to other sources of uncertainty discussed here. This may be indicative that the structure of QCD and Top jets are distinct enough, and the CNN is robust enough to distinguish between them even when blurred.

Finally, one should be aware that the performance of the CNN (or any ML method) can change each time it is run due to the training procedure and how data is sampled. This has the potential to affect the LLR obtained. As mentioned in Sec. 2.2.1 and shown in Fig. 4, we have accounted for this effect by Bootstrapping and found that the significance change over different trainings was subleading respect to the cross-section uncertainties considered above.

## 2.3 Use case II: Higgs EFT kinematics

Now we will perform the same simple hypothesis test, but this time using as input multiple high-level kinematic features, instead of single images per event. We then move into another physics context, namely searches for new phenomena within the Effective Field Theory (EFT) framework.

Within the EFT, effects of new physics are encoded as an expansion in momenta, which leads to modifications of the SM particles' behaviour following this structure,

$$\mathcal{L}_{EFT} = \mathcal{L}_{SM} + \mathcal{L}_{BSM}, \quad \text{where} \quad \mathcal{L}_{BSM} = \frac{1}{\Lambda^{2n}} \sum_i c_i \mathcal{O}_i. \tag{6}$$

Here $\mathcal{O}_i$ represents an operator made from SM fields and singlet under all SM symmetries, $c_i$ is the corresponding Wilson coefficient and $\Lambda$ is the scale of new physics.

To illustrate the use of ML methods and their translation into statistical significance, we will choose a well-studied final state in the EFT context,

$$p\,p \to Z^* \to h\,Z, \quad \text{where} \quad Z \to \ell^+\ell^- \text{ and } h \to b\bar{b}, \tag{7}$$

---

[5]The filter was applied using the scikit-image libraries in python [35], https://tinyurl.com/gaussianfilter.

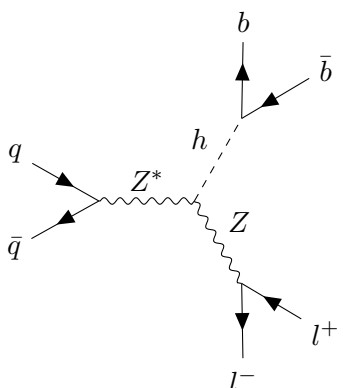

Figure 9: Associated production of a Higgs with a massive $Z$ boson in the 2-lepton final state.

also shown in Fig. 9.

EFT effects appear at the level of production of the $Zh$ final state and, to a lesser extent, in the decay products. Here we will choose a modification of the $hZZ$ coupling coming from one of the possible operators $\mathcal{O}$ in Eq. 6 at leading-order,

$$\mathcal{L}_{BSM} \supset ig \frac{c_{HW}}{m_W^2} (D^\mu H)^\dagger \sigma_a (D^\nu H) W_{\mu\nu}^a, \tag{8}$$

where we follow the notation in Ref. [36]. $H$ denotes the Higgs doublet, where the field $h$ resides, and $W_{\mu\nu}$ is the $SU(2)_L$ field strength.

After electroweak symmetry breaking, this term will affect the Higgs coupling to $Z$ bosons, producing a modification of the overall cross section. This modification can be expressed as a quadratic polynomial in the BSM parameter $c_{HW}$,

$$\sigma_{EFT} = \sigma_{SM} + c_{HW}\,\sigma_{int.SM-BSM} + c_{HW}^2\,\sigma_{pureBSM}. \tag{9}$$

In the limit $c_{HW} = 0$ we recover the SM result. The linear term in $c_{HW}$ is an interference term between the SM production and the new EFT effects. Finally, the quadratic term in $c_{HW}$ represents the pure BSM cross-section.

The EFT will also modify the kinematic features of this final state [36, 37], which we use in the classification task. In Appendix A we provide more details on the kinematic features considered in this study, along the lines of what was already studied in Ref. [38]. Here we will use 13 parton-level kinematic variables as inputs for the ML algorithm.

Contrary to the *Use Case I: Top vs QCD* in the previous section 2.2, here we cannot sensibly produce synthetic data to train on *pure* classes. What we need to consider instead is a pure SM class, and a New Physics class which contains the SM as part of the kinematics entangled by the interference term. This is seen explicitly in Eq. 9, where the EFT is a mixture of SM and BSM effects.

To provide results, we choose to perform a test between SM only events (equivalent to $c_{HW} = 0$) and EFT = SM + BSM events with a small value $c_{HW} = 0.005$ [39]. In other words we want to test $H_0 = H(c_{HW} = 0)$ against $H_1 = H(c_{HW} = 0.005)$.

In this section we will assume the background $b$ is a well known value, and the signal $s$ is taken to be the difference between the observed number of events and the background. Over a large number of experiments, the average of this signal would approach the true (or theoretical if the theory is correct) value.

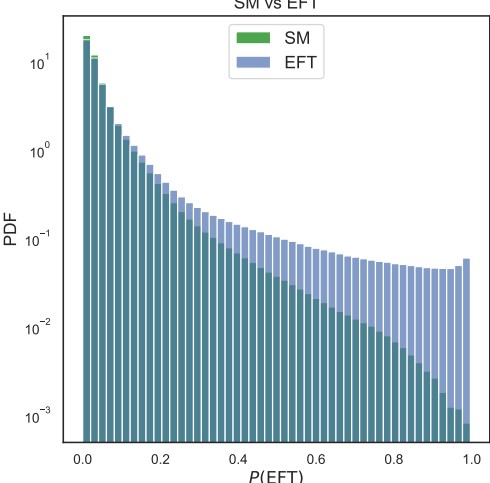
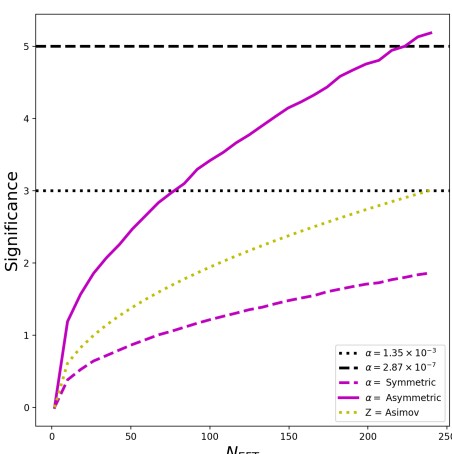

Figure 10: *Left plot:* PDF distribution of the output of the classification task, P(EFT), for samples with pure SM events and with EFT ($c_{HW} = 0.005$) events (the EFT manifestly including SM events). *Right plot:* The significance level of symmetric and asymmetric tests for parton-level kinematics classifier, expressed in terms of number of standard deviations of the Gaussian for no cut in $P(\text{EFT})$. Standard discovery significances are shown for comparison.

We train a Deep Neural Network (DNN) to classify events as either SM or EFT by training on pure SM and EFT events generated in `MadGraph` [36] (with no showering and detector effects) in a similar fashion as we did with the CNN before, except now we use a DNN taking parton-level kinematic data as input rather than a CNN taking detector-level jet images. The DNN is better suited for higher dimensional data than a CNN, and the aim here is to show that we can construct a good hypothesis test regardless of the ML method.

The DNN is trained using truth labels and optimised for the data at hand. It outputs a predicted probability which we translate into the predicted probability $P(\text{EFT})$ of any given event belonging to an EFT distribution.

The classifier output $P(\text{EFT})$ is shown in the left panel of Figure 10. Using this classifier output as a PDF, we follow the same procedure as before; sampling a number of events to find Likelihoods, and finding the LLR. We then apply the symmetric and asymmetric conditions to find the significance level of the test, which we translate into $n_\sigma$ as shown in the right panel in Figure 10. The improvement from the Asimov significance (yellow-dashed) to the significance obtained by including ML outputs (purple solid) is an expected result. Note also that we show results with no cut on $P(\text{EFT})$, since we have already discussed the effects of the cut in the previous case, and there cutting gives a small improvement before falling off due to having low number of total events.

## 3 Unsupervised Machine Learning and Generalised hypothesis testing

Although the use of Machine Learning in Particle Physics is dominated by classification problems, exciting developments in the field of anomaly detection using unsupervised ML techniques are quickly advancing. These techniques have the potential to detect new signals without any prior knowledge of them. Whilst the concept of anomaly detection is not unique to

ML, see for instance [40–43], ML methods offer much promise and huge potential [44] to facilitate a discovery.

In the context of this paper, we will explain how to perform a hypothesis test using the outputs of unsupervised ML, which we will showcase for a Variational Autoencoder (VAE) trained to detect anomalous events from a SM background.

In the language of hypothesis testing, anomaly detection translates into assuming that the background (null) is well known, however the signal (alternative) is not. Here we shall outline how a generalised hypothesis test can be performed in such a case.

## 3.1 Composite hypothesis testing

As with simple hypothesis testing, the generalised test is usually performed using a Likelihood Ratio but now the values of the parameters $\boldsymbol{\theta_0}$ and $\boldsymbol{\theta_1}$ are unspecified. That is to say, generally data $X$ is observed, which we say to be sampled from one of two PDFs $p_0$ and $p_1$ given two general hypotheses:

$$H_0 : X \sim p_0(x|\boldsymbol{\theta}_0) : \boldsymbol{\theta}_0 \in \Theta_0,$$
$$H_1 : X \sim p_1(x|\boldsymbol{\theta}_1) : \boldsymbol{\theta}_1 \in \Theta_1,$$

where the sets $\Theta_0$ and $\Theta_1$ represent all possible values for the parameters.

In our case the null hypothesis is simple so there will be only one possible value for $\boldsymbol{\theta_0}$ but the alternative hypothesis is composite so $\boldsymbol{\theta_1}$ could still be any possible value, representing possibilities for new physics.

The generalised Likelihood Ratio is

$$\lambda(\boldsymbol{\theta} \mid H_i) = \frac{\max_{\boldsymbol{\theta}_0 \in \Theta_0} L(\boldsymbol{\theta}|H_i)}{\max_{\boldsymbol{\theta}_1 \in \Theta_1} L(\boldsymbol{\theta}|H_i)}. \tag{10}$$

We can say that $\hat{\boldsymbol{\theta}}_0$ is the value of $\boldsymbol{\theta}_0$ which maximises the Likelihood under $H_0$, called the Conditional or Restricted Maximum Likelihood Estimate, and $\hat{\boldsymbol{\theta}}_1$ is the Maximum Likelihood Estimate of $\boldsymbol{\theta}_1$. We will see that $\hat{\boldsymbol{\theta}}_0$ will be taken as the known value for the background and $\hat{\boldsymbol{\theta}}_1$ will be the observed quantity to test against the background. We are left with

$$\lambda(\boldsymbol{\theta} \mid H_i) = \frac{L(\hat{\boldsymbol{\theta}}_0 \mid H_i)}{L(\hat{\boldsymbol{\theta}}_1 \mid H_i)}. \tag{11}$$

### 3.1.1 Generalised hypothesis testing using VAE outputs

Applying this test to our scenario, we must be careful in considering what is the parameter of our model. In a counting experiment the parameter of interest would be the mean number of signal events $s$ [6] but our model also contains a term describing the Likelihood of obtaining a given value for the ML output. In particular, for a VAE we will consider the Reconstruction Error $R$ [7] as the PDF parameter and exemplify its use in the context of the EFT described in the previous section. See section 3.2 for more details.

---

[6]Another parameter of the model could be the mean number of background events $b$ - this is either treated as a known quantity or as a nuisance parameter, the latter being done in [34].

[7]Although the Reconstruction Error is an obvious choice for output, one could think on other ways to represent the VAE output which could serve as a PDF. For example, one could use information in the latent space to do clustering and building some distance on that space. Also a non-ML alternative would be the Mahalanobis distance [45], which is a measure of how far a data point is from the center of a distribution - there usually a PCA is performed on data and the Mahalanobis distance calculated on the resulting distribution.

First, we write our Likelihood as

$$L(s, c_{HW} \mid H(c_{HW})) = \frac{(s+b)^n \exp(-(s+b))}{n!} \cdot \prod_{k=1}^{n} p(R_k \mid H(c_{HW})), \qquad (12)$$

where we are now saying that the possible hypotheses can be any number of hypotheses for a given value of the parameter $c_{HW}$ - note that the SM can be described by the hypothesis with $c_{HW} = 0$; this fact will allow us to test between the SM and any hypothesis whose $c_{HW}$ parameter is unknown (and potentially even other *forms* of hypotheses if the form of the PDF is similar!). The Poisson term gives the dependence on $s$ whereas the second term from the VAE output depends on $c_{HW}$ which defines the signal. Notice that we need not write the dependence on $b$ explicitly since we take it to be known.

The Likelihood is allowed to depend on any number of parameters although we will see later that the number of parameters will affect the test statistic LLR distribution.[8] We can actually say however that $s = \mathcal{L}_{\text{int}} \sigma_{\text{EFT}}(c_{HW})$ with $\mathcal{L}_{\text{int}}$ integrated luminosity. And since, in this case, there is a strict dependence of the signal cross-section on $c_{HW}$, then our Likelihood will depend on just one parameter, which could be either $s$ or $c_{HW}$.

Be aware that this result may not be applicable to every signal and one should consider the number of parameters of interest carefully for each unique experiment which would yield their own Conditional Maximum Likelihood Estimate for the Likelihood. Also in principle the Likelihood may contain a number of nuisance parameters such as the aforementioned background mean $b$ or even the integrated luminosity $\mathcal{L}_{\text{int}}$ if they are not well known. In our setup we shall assume that they are well known and will not be considered nuisance parameters. Note that we have also previously shown in Sec. 2.2.4 an alternative method for accounting for uncertainties in the background.

Given all this, the Likelihood Ratio which compares some measured value $\hat{s}$ to some value $s$ under the null can be expressed as

$$\lambda(s, \hat{s} \mid H_i) = \frac{L(s \mid H_i)}{L(\hat{s} \mid H_i)}. \qquad (13)$$

Remember here that we obtain the Likelihood Ratio *given* some hypothesis is true, but (without truth information) we do not know which hypothesis is true.

We are interested in discovery, so we shall want to compare the observed value $\hat{s}$ to $s = 0$. We will use the test statistic

$$q_0(\hat{s} \mid H_i) = \begin{cases} -2\ln \dfrac{L(0 \mid H_i)}{L(\hat{s} \mid H_i)} & \text{for } \hat{s} \geq 0, \\ 0 & \text{for } \hat{s} < 0. \end{cases} \qquad (14)$$

Here one should remember that $\hat{s}$ is the observed value which would be distributed around some unknown mean $s'$. Whilst in an normal well set-up experiment we cannot have negative mean number of signal events[9] we still write the condition for $\hat{s} < 0$ since in some experiments the number of events could be observed to be less than the expected background simply because they values are Poisson or Gaussian distributed but we know that $\hat{s} < 0$ would not align with our hypothesis. Also observe that if the observed data happened to be the true mean $s' = 0$ then one would obtain $q_0 = 0$.

We will want to obtain the distribution of the test statistic under which we can find the significance level $\alpha$. However there are a few important points to mention here. The first is

---

[8]Note that we shall use 'test statistic' rather than 'LLR' to keep in line with other literature but the two are interchangeable here.

[9]Although this could be obtained due to some systematic error or in an experiment such as neutrino oscillation detection where the signal hypothesis may result in fewer events than the background.

that since we do not know the alternative hypothesis, we cannot find the distribution of the test statistic under it and hence cannot find $\beta$. The second is as another consequence of not knowing the alternative hypothesis, we cannot even make any reliable claims on what the best $\alpha$ to use even is since the average observed test statistic value would change under different hypotheses. This may seem like it would be a problem for when we want to perform our test; however it simply means that there is no reasonable way to find $\alpha$ based on the alternative hypothesis as we did before (this test therefore has more in common with Fisher's methods). Whilst there is much merit to finding some optimal $\alpha$ e.g. through the symmetric condition or ROC curves as in simple hypothesis testing, it is not required. All one requires to set the claim one hypothesis in favour of the other is to obtain some observed test statistic that is above below some significance level cutoff. One can set $\alpha$ before an experiment e.g. as 0.05 or $2.87 \times 10^{-7}$ and compare the $p$-value obtained to that. Therefore what we shall outline below is how to find the $p$-value and significance $Z$.

### 3.1.2 Obtaining p-values

We can find the area which is the $p$-value under the tail of the $q_0$ distribution. Given some measured value $\hat{s}$, we will compare the resultant $q_{0,\text{obs}}$ to the distribution $f(q_0(\hat{s} \mid H_0))$, that is the distribution of $q_0$ assuming that the null is true. Or in other words: assuming that the null is true and we obtain some data from an experiment, what is the probability that our data were not obtained from the null? If there was some sizeable deviation in our obtained data from the background hypothesis, then the probability of the background being true would be small and thus an indication that real signal is present.

But what is the distribution $f(q_0(\hat{s} \mid H_0))$? One could find it with costly Monte Carlo simulations with different $\hat{s}$, with $\hat{s}$ having expected value $s' = 0$. Fortunately one can use the brilliant theorem from Wilks and Wald that the test statistic is chi-square distributed [46,47].[10] Given the our proper construction of the hypotheses, we only require that we have sufficient data in order for Wilks theorem to hold. Going by the results of Ref. [34] we should meet that criteria. See Ref. [48] for further discussion on the context and applicability of Wilks theorem.

Let us use the shorthand $q_0 \equiv q_0(\hat{s} \mid H_0)$ from now on. Since we have one parameter of interest, we will have a chi-square distribution with one degree of freedom. To account for the fact that our test statistic is zero for any non-physical $\hat{s} < 0$ (half of all events) $q_0$ will follow a half-$\chi_1^2$ distribution:

$$f(q_0) = \frac{1}{2}\delta(q_0) + \frac{1}{2}\frac{1}{\sqrt{2\pi}}\frac{1}{\sqrt{q_0}}\exp\left(-\frac{q_0}{2}\right). \tag{15}$$

This is the distribution of the test statistic assuming that the null is true - if in an experiment one measures some value $q_{0,\text{obs}}$ then the $p$-value is found by

$$p\text{-value} = \int_{q_{0,\text{obs}}}^{\infty} f(q_0)\,\mathrm{d}q_0. \tag{16}$$

We obtain a value for $q_{0,\text{obs}}$ with $\hat{s} = n - b$ which we take to be the average value that would be obtained; i.e. $\hat{s} = \mathcal{L}_{\text{int}}\sigma_{\text{EFT}}^{\text{pure}}(c_{HW} = 0.005)$ then find the $p$-value using equation (16) using the distribution in equation (15) and convert to a significance using equation (20), the results of which we will show in the next section, Figure 12.

---

[10]This theorem formally it states: assuming that the distribution of some observable $x$ in general depends on $m$ unknown parameters, and under $H_0$ it depends on $m_0$ unknown parameters, then if the null is true, and under some regularity conditions, then the test statistic $q_0$ converges to a $\chi_\nu^2$ distribution, with $\nu = m - m_0$.

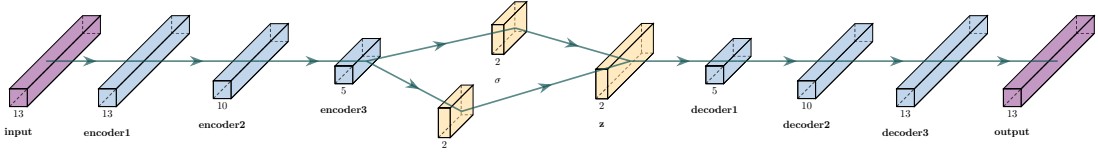

Figure 11: The VAE used. The data input consists of 13 dimensions which are fed into the encoder with three layers of 13,10 and 5 dimensions. The compressed information is split into parts, each with 2 dimensions which together comprise the latent space vector which is then fed into a decoder with three layers of 5,10 and 13 dimensions to yield an output of 13 dimensions. All layers use a ReLU activation function, except for the last which uses a sigmoid.

## 3.2 Use case: VAE output for EFTs

Autoencoders and Variational Autoencoders has emerged as potent tools for dimensionality reduction, de-noising images and generation of new unseen data [49, 50] as a result of their ability to learn a model distribution of data and to create a model able reconstruct it. They have also been studied in the context of anomaly detection, see e.g. [51–53], whereby they are trained on only background data such that when fed data which deviates from the norm they are able to provide an indication of it being potentially anomalous.

Generally speaking, a VAE consists of a *probabilistic encoder* $q_\phi(z|x)$ which takes the input vector $x$ to a latent vector space $z$. A *probabilistic decoder* $p_\theta(x'|z)$ takes the latent vector to a reconstructed output $x'$. Here $\phi$ and $\theta$ denote parameters in the two networks.

A VAE is trained by minimising the Kullback-Liebler divergence between $q_\phi(z|x)$ and $p_\theta(x'|z)$ as well as Log-Likelihood term, hence obtaining the optimal $q_\phi(z|x)$ for encoding the input data and $p_\theta(x'|z)$ for taking the latent space representation of the data into a reconstructed output, for data of the same type as the training data.

When given data different to the training data (which can include more spurious background data), $x'$ will be less similar to $x$ than for data typically found in training. We will make use of the Reconstruction Error

$$R = \left| x' - x \right|^2, \text{ where } x' = p_\theta\left(q_\phi(x)\right),$$

to quantify this difference between input and output. We expect that the Reconstruction Error is larger for more anomalous images.

If we were performing an event-by-event anomaly detection analysis, we could simply choose a threshold value for the Reconstruction Error above which events are classified as anomalous, and display the results in a ROC curve. However we have no need to do so when we perform a hypothesis test, rather we only need to construct a PDF of the Reconstruction Error without choosing any threshold value. It may still be useful to use one, however, to trim the PDF data to include a higher proportion of signal candidates, but we are just interested in demonstrating the testing process here.

At any rate, it is still useful to plot a ROC curve when building the VAE, since we can adjust its architecture to maximise the AUC. Based on this procedure, we use an encoder with three layers of 13, 10, 5 dimensions, a latent space in two dimensions and a decoder with three layers of 5, 10, 13 dimensions as shown in Figure 11. The input and output layers both have dimensions of 13 to match the number of parton-level variables.

However we have no need to pick a threshold value when we perform a hypothesis test, rather we only need to construct a PDF of the Reconstruction Error. Once the VAE is optimised,

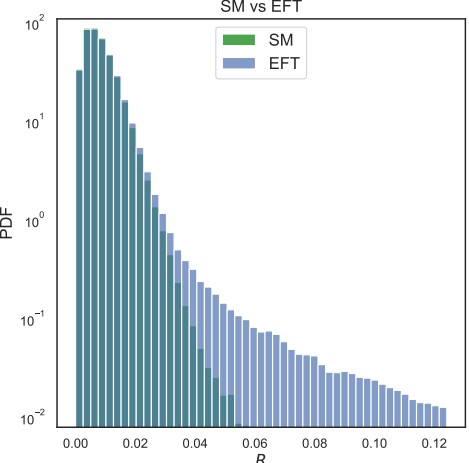
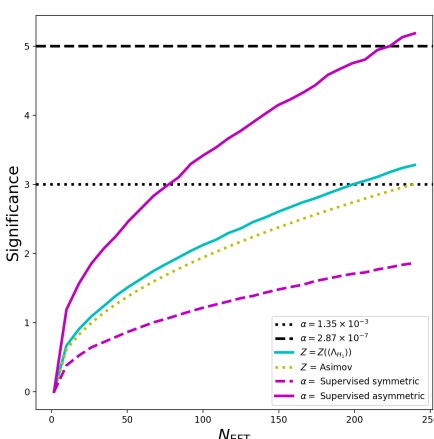

Figure 12: *Left plot:* PDF distribution of the output of the classification task, Reconstruction Error $R$, for samples with pure SM events and with EFT ($c_{HW} = 0.005$) events (the EFT manifestly including SM events). *Right plot:* The significance level from a generalised Likelihood Ratio Test for EFT anomaly detection, expressed in terms of number of standard deviations of the Gaussian.

we construct our PDF of the reconstruction error for a dataset containing a number of pure SM events which shall serve as our background PDF to compare against, and also for a dataset containing EFT events, the same data being used as in the supervised case, which we shall use as our observed dataset.

We show in the left panel of Figure 12 the two distributions. The goal will be to obtain a measure of the significance of our observed dataset containing signal. Note that without the truth information we would not know what type of signal this were, only that it is indicative of something not in the SM dataset.

Since we will not have an alternative distribution to compare against we will not be able to find the significance level as we did before. Instead, as detailed in the previous section 3.1.2, we can only quote the *p*-value which can be compared to some manually fixed significance level, rather than one tailored to a specific experiment. The results of this procedure are shown in the right panel of Fig. 12. In this plot we compare the VAE composite testing (turquoise solid line), the previous results from the supervised task (purple solid and dashed lines) and the Asimov (yellow) significances as a function of the number of EFT events.

One would expect that performing a simple hypothesis test with a VAE would give worse results than performing a simple hypothesis test with the DNN, since the DNN is a supervised algorithm and the VAE is not. Moreover, we would also expect that a simple hypothesis test with the VAE would perform better than a generalised hypothesis test with the VAE. We find both these points to be true, with the significance from the simple hypothesis test with the VAE being slightly higher than the significance from the generalised hypothesis test with the VAE, but lower than the simple hypothesis test with the DNN.

## 4 Conclusions

In this paper we proposed various ways to incorporate the outputs of Machine Learning supervised and unsupervised tasks into the study of statistical significance.

We first described the method for a supervised task with two different types of searches, one

where information was expressed as an image and another where it was introduced as a set of tabular features. Firstly, we considered an image representation of events with hard hadronic activity, thoroughly studied in the area of top-tagging. Then we looked into a typical channel for SMEFT searches, the associated production of the Higgs with a massive vector boson. In this second case, the inputs for the ML were high-level high dimensionality kinematic features.

In both use cases, we showed how from a ML output one could perform a simple hypothesis testing. In one case, the test was distinguishing between a "pure QCD" hypothesis and a "QCD with Top" hypothesis. In the second example, the test was to distinguishing between a "SM only" hypothesis and a "SM with EFT effects" hypothesis for $ZH$ kinematics.

We expressed the test statistic as significance using different methods. We showed how to implement a symmetric and an asymmetric testing condition, based on computing the areas under the Log-Likelihood Ratio distributions found from a number of toy experiments. These areas can be used to determine the significance level $\alpha$ for simple hypotheses. We also showed the number of signal events required for these conditions to match the fixed value of $\alpha$ typically used for discovery.

As expected, we found that by incorporating ML outputs into a Log-Likelihood ratio we obtain a stronger hypothesis test, leading to better significance, and quantified this gain by comparing with the Asimov significance.

For unsupervised tasks, we proposed a generalised Likelihood Ratio Test, which can be performed to test between a known SM background and an unknown signal. We discussed options for hypothesis testing tailored to this case, and present results for a generalized hypothesis testing where the New Physics is treated as a composite hypothesis. We incorporated the PDF of the output from a VAE (Reconstruction Error) into the Likelihood Ratio, and obtained a discovery significance $Z$ for an average experiment by computing the area under the Log-Likelihood Ratio, which we assumed to be chi-square distributed via Wilks theorem. We again find an improvement in the significance when incorporating the VAE output.

Finally, let us mention that in this paper we have focused on relatively simple ML tasks, but the proposal of adding the ML output as part of the likelihood as in Eqs. 3 and 12 could be adapted to more complicated situations. For example, one could consider the addition of ML supervised tasks as described in Sec. 2.3 in global fits to SM Effective Field Theory (SMEFT) [39,54–66], where some channels like $VH$ [38], Vector boson fusion [67] or di-Higgs could greatly benefit from adding information on ML training in simulated events. Moreover, future public releases of likelihoods from experiments [68] could incorporate information on ML training outputs following the lines of this paper. Also, it would be interesting to undertake a Bayesian model selection analysis [25,69], namely finding the Bayes factor from the PDFs, although doing so would require answering many more questions regarding the parameters and priors [70] and how to implement them in noisy classifications, e.g. [71].

**GitHub code:** Note that the implementation of the calculation of hypothesis testing can be found in the `GitHub` repository:
`https://github.com/high-energy-physics-ml/hypothesis-testing`.

**Funding information** VS is supported by the PROMETEO/2021/083 from Generalitat Valenciana, and by PID2020-113644GB-I00 from the Spanish Ministerio de Ciencia e Innovación. MS acknowledges support by the Data Intensive Science Center in the South East Physics Network (DISCnet), an extension of the STFC, under grant number ST/P006760/1.

# A Details on event generation and data selection

We consider two different datasets for our analyses. The first dataset is intended as an example of a preexisting search for the discovery of an expected particle, namely the Top quark against a QCD background. This is not a search for new physics, but rather an example of how the Top quark could have been discovered with Machine Learning prior to its 1995 discovery. This search will showcase our methods in a setting where the data is well understood, so as to focus on the methods themselves before moving on to new searches.

The second dataset we consider will be of associated production of the Higgs and a massive vector boson, where effect of new physics modify the production in the context of the Effective Field Theory (EFT).

## A.1 Fat Jets

We considered boosted hadronic activity where the leading jet $p_T$ is very high (more than 750 GeV). The QCD dijet and hadronically decaying $t\bar{t}$ processes are simulated using `Madgraph 5` [72] and `Pythia 8` [73]. All the processes are simulated for the LHC proton proton collisions at $\sqrt{s}$=13 TeV. In case of $t\bar{t}$, we use Madspin [74] for the decay. After parton-level simulation and hadronization, we cluster the jets using an Anti-kt algorithm with R=1.

The cross-sections after accounting for efficiencies are $\sigma_{QCD}$=48829.2 fb, $\sigma_{top}$=53.1684 fb.

We produce grey scale $p_T$ weighted calorimeter images ($\eta$ and $\phi$) of the leading jet constituents. The $\eta, \phi$ resolution is 0.087 and 5 degrees, respectively. The centre of the images is aligned with the jet centre, and then image size is set to 25 × 25. Images are zero-padded in case the original images had fewer pixels. In Fig. 13 we show the averaged images for both the cases.

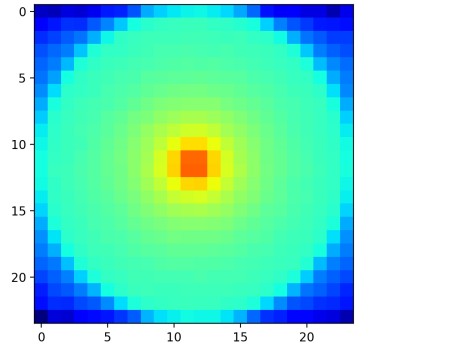 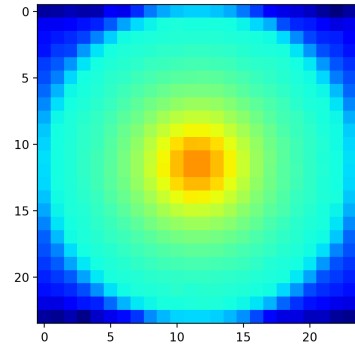

Figure 13: Representation of the average QCD (left) and Top (right) events.

## A.2 EFT effects in ZH associated production

We consider the ZH Channel in the EFT set-up. We also switch on a single EFT operator, represeented by the $c_{HW}$ coefficient. In this channel, we consider the 2L option, where the Higgs decays to $b\bar{b}$, and Z to a pair of leptons.

The HEFT model [36] is used in the `UFO format` [75] in `Madgraph` [72]. For consistency, background events (i.e. SM) are also simulated using the same model, setting zero values of all EFT coefficients. The cross-sections, after taking into account efficiencies to the cuts shown in Table 1, are $\sigma_{SM} = 14.0$ fb, $\sigma_{EFT}(c_{HW} = 0.005) = 17.1$ fb.

Based on these simulated events, we construct the following kinematic variables :

Table 1: Generation level cuts applied for both signal and background processes.

| Particles | Cuts |
|---|---|
| b partons | $p_T^b > 20$ GeV, $|\eta_b| < 2.5$, |
| | Leading b-jet $p_T > 45$ GeV |
| leptons | $p_T^l > 7$ GeV, $|\eta_l| < 2.7$, $p_T^V > 75$ GeV |
| | Leading lepton $p_T > 27$ GeV |

1. $p_T^{b_1}$: transverse momentum of the leading b parton.

2. $p_T^{b_2}$: transverse momentum of the sub-leading b parton.

3. $p_T^{l_1}$: transverse momentum of the leading lepton.

4. $p_T^{l2}$: transverse momentum of the sub-leading lepton.

5. $p_T^H$: transverse momentum of the Higgs.

6. $\eta_H$: pseudorapidity of Higgs.

7. $\phi_H$: azimuthal angle of Higgs.

8. $\delta R_{l_1 l_2}$: angular separation between leptons.

9. $\delta R_{b_1 l_1}$: angular separation between leading b and leading lepton.

10. $M_T^{ZH}$: transverse mass of ZH.

11. $p_T^{ZH}$: transverse momentum of the leading b parton.

12. $\delta \phi_{l1b1}$: azimuthal separation between leading b and leading lepton.

13. $d\phi_{l1b2}$: azimuthal separation between sub-leading b and leading lepton.

# B  Significance levels of tests

When it comes to choosing the significance level $\alpha$ one has some freedom. Actually one has as much freedom as they like, however the smallest feasible values are typically best. We shall explore two sensible options, a symmetric method and an asymmetric method to determine $\alpha$. We show these with the intent to display how the distinction between two hypotheses changes with the number of observed events.

These options could be directly applied to experiments attempting to distinguish between two relatively equally credible hypotheses e.g. distinguishing between Higgs spin hypotheses [32], however for discovery against a background one might wish to use a set stronger criteria. Indeed this is what is typically done for discovery; the standard 5-$\sigma$ in HEP is to set $\alpha = 2.87 \times 10^{-7}$ whereas the 3-$\sigma$ for evidence corresponds to $\alpha = 1.35 \times 10^{-3}$.

We shall showcase these two options in the context of Top discovery and EFT effects as it is useful to know the results one would get *if* they were to use them and also because the asymmetric condition is actually, on average, what one would obtain for the significance obtained from the *p*-value.

However, in general there is no set *correct* $\alpha$ to use. One may also consider a ROC curve of $\alpha$ and $\beta$ (to see how to find this see Appendix A of [69]) to find some optimal value for a specific experiment. This is where one's preferences for the importance of type I errors against type II errors and understanding of the two hypotheses would come into play.

## B.1 Symmetric testing condition

We first consider the scenario where we want the probability of type I error to be equal to the probability of type II error i.e. the two hypotheses are treated equally. Ergo once obtaining the LLR distributions as in Figure 5 we find $\alpha$ as the area under the right-side tail of $f(\Lambda_{H_0})$ and $\beta$ as the area under the left-side tail of $f(\Lambda_{H_1})$, using

$$\alpha = \frac{\int_{\Lambda_{\text{cut}}}^{\infty} f_0(\Lambda) \mathrm{d}\Lambda}{\int_{-\infty}^{\infty} f_0(\Lambda) \mathrm{d}\Lambda} \tag{17}$$

and

$$\beta = \frac{\int_{-\infty}^{\Lambda_{\text{cut}}} f_1(\Lambda) \mathrm{d}\Lambda}{\int_{-\infty}^{\infty} f_1(\Lambda) \mathrm{d}\Lambda}, \tag{18}$$

and finding $\Lambda_{\text{cut}}$ such that $\alpha = \beta$. We can convert the $\alpha$ obtained into a number of standard deviations $n_\sigma$ of the (one-sided) unit Gaussian by solving

$$\alpha = \frac{1}{\sqrt{2\pi}} \int_{n_\sigma}^{\infty} \exp\left(\frac{-x^2}{2}\right) \mathrm{d}x. \tag{19}$$

Note that this does *not* require that the LLR be Gaussian distributed,[11] but we will assume so for communication purposes, as quoting a very small value of $\alpha$ becomes unwieldy very quickly. For example $\alpha = 0.05$ corresponds to $n_\sigma = 1.64$ and a 5-$\sigma$ discovery corresponds to $\alpha = 2.87 \times 10^{-7}$. This can be seen in the plots in Figure 6. We may increase the separation of the distributions by taking some threshold $P_{\text{cut}}(\text{Top})$ that reduces the number of events sampled from. The reasoning behind this is that in doing so we would consider a relatively larger amount of signal events than background events as shown in the right plot in Figure 3. We actually find that if we apply this cut to both terms in the Likelihood the separation actually decreases. This seems counter-intuitive at first but makes sense when it is realised that the whilst there will more relatively more Top events than QCD events after the cut than before (although still less overall) and a higher chance of sampling a Top, the most important aspect of the ML term of the Likelihood are the points where the PDF is highest i.e. the lowest values of $P(\text{Top})$ - where the difference between QCD and QCD + Top PDFs is greatest (remember that Figure 3 is in log-scale). By excluding these values the Likelihood Ratios are actually less separated. However we can still use the same cut threshold to alter the number of events in the Poisson term (hence making the most of both terms and still using the output of the CNN for making this cut). When making such a cut we see that the separation significance increases as we increase this cut threshold (up until around 0.8 $P_{\text{cut}}$, after which the number of events left after the cut becomes low enough to reduce separation again) as shown in Figure 6. From this we can see the number of events required for this method to reach the same strictness as $\alpha = 2.87 \times 10^{-7}$, illustrating when it may or may not be appropriate to use these conditions if one is looking for discovery. The separation significance $n_\sigma$ from this condition is always lower than that of $\alpha = 2.87 \times 10^{-7}$ in the ranges shown, however the symmetric condition is

---

[11]This is indeed our case, and for all simple hypothesis test with sufficient data but we still use this quantity even when the LLR is non-Gaussian.

has more than the asymmetric one and both conditions do get to the same significance level as $\alpha = 2.87 \times 10^{-7}$ given enough events.

We show also in these figures some common *discovery* significances used in counting experiments; $s/\sqrt{b}$, $s/\sqrt{s+b}$ and the Asimov significance. These are actually the average of the significances that would be obtained from the $p$-value (see sec B.3) after an experiment and are not exactly the same as $n_\sigma$, however we can still compare them, as they will on average have the same value as the asymmetric $n_\sigma$ if one considered only the Poisson factor in the Likelihood. We therefore show both the *separation* significances from the symmetric and asymettric hypothesis tests alongside these discovery significances in the figure (and subsequent figures) with the general $y$-axis label of 'Significance' being appropriate for both. These discovery significances arise from of the Poisson term in the Likelihood finding a $p$-value as shown in Appendix B.4 as well as evidence of the previous statement.

## B.2  Asymmetric testing condition

We also consider another condition where one is favours accepting or rejecting the null without consideration of the type II error. A test using such a condition could be performed in cases where one is more concerned about finding whether or not data agrees with the null than if the data actually suggests that the alternative is a good hypothesis. Such an approach draws from the reasoning of Fisher [76] who was famously opposed to formulating an alternative hypothesis, although here we must still know the form of the alternative in order to find the LLR. Such an approach can be called a Fisher/Neyman hybrid (although both Fisher and Neyman may both have disagreed with calling it such!).

Here we find $\alpha$ as the area under from tail of the LLR distribution under the null from $\Lambda_{\text{cut}} = \langle f_1(\Lambda) \rangle$ to infinity: hence we use equation (17) with this new cutoff value. As before we can convert this into $n_\sigma$ which we show in Figure 6. Only with a cut of 0.8 $P_{\text{cut}}$ does this condition reach the same strictness as $\alpha = 2.87 \times 10^{-7}$.

It is actually the case that the $n_\sigma$ obtained under this condition will be the same as the average significance $Z$ obtained from the $p$-value of an experiment. Therefore in understanding this method, one should have all the tools required to obtain a $p$-value and report which hypothesis their experiment favours. In the next section we shall explain the $p$-value in more detail.

## B.3  $p$-values and reporting of results after obtaining data

We have so far outlined two procedures for finding the significance level $\alpha$ of a simple hypothesis test (which we translate into a separation significance $n_\sigma$) by using toy experiments to obtain LLR distributions. What then would the procedure for accepting or rejecting either hypothesis look like in practice when one obtains data from a real experiment and how could one quantify the extent to which observed data agrees or disagrees with the null? The answer to the first part of this question is simply to find the LLR value obtained given observed data and see whether or not it is above or below the cutoff value $\Lambda_{\text{cut}}$ that defines the significance level. One would then say whether the data favours the null or the alternative. Remember that it cannot be said that data *is proving* one hypothesis or the other, only that in that experiment, out of the two hypotheses tested the data agrees more with one than the other.

The answer to the second part of the question would to to report the $p-value$, that is the area under the tail of the LLR distribution which assumes the null to be true from the observed LLR value to infinity. This is just another way of saying the commonly given definition "The $p$-value is the probability under the null of obtaining data as extreme or more extreme than the observed data" [1]. The $p$-value is often translated into a number of standard deviations of the one-sided unit Gaussian in the same way as equation (19), but we shall write it again

here to avoid confusion:

$$p-\text{value} = \frac{1}{\sqrt{2\pi}} \int_Z^\infty \exp\left(\frac{-x^2}{2}\right) dx. \tag{20}$$

Here one solves for the significance $Z$ which is confusingly called the significance, not to be confused with the significance level, which can be set prior to an experiment; the $p$-value and subsequent significance $Z$ are obtained only after an experiment. The $p$-value will also change with each repeat of the experiment as differently randomly sampled data results in different LLR values. We note however that on average, the discovery significance $Z$ would have the same value as the separation significance $n_\sigma$ under the asymmetric condition.

Finally as well as $p$-values or $Z$, one may also report Conditional Error Probabilities (CEPs). For frequentists they correspond to the type I and type II error probabilities We will not explore them here but [77] and [78] give good discussions on them.

### B.4 Common discovery significance definitions and their relationship to Log-Likelihood ratios

Commonly used discovery significances are $s/\sqrt{b}$, $s/\sqrt{s+b}$ and the Asimov significance $\sqrt{2((s+b)\ln(1+s/b)-s)}$. It is important to note that these are the average (median) significances that one would obtain from computing the average significance over a number of different experiments, although when the expected numbers of signal and background events are well known, then they can be used directly and reliably. The former is considers there to be no uncertainty on $b$ and related to the Asimov significance when $s \ll b$. Although they can be derived from more heuristic methods, these actually arise from considering only the Poisson term in the Likelihood (i.e. are most applicable in any counting experiment). It is interesting that the first two can be organically shown in the case of a simple hypothesis test whilst the Asimov significance can be shown in the case of a generalised hypothesis test; the first two being limiting cases of Asimov significances with and without uncertainty on $s$.

Let us see how the significances $s/\sqrt{b}$ and $s/\sqrt{s+b}$ can be found from the ratio of Poisson Likelihoods for a simple hypothesis test. If we take

$$\Lambda = -2\ln\left(\frac{\text{Pois}(b|n)}{\text{Pois}(s+b|n)}\right) = 2\left(n\ln\left(\frac{s+b}{b}\right)-s\right), \tag{21}$$

then we can find the number of standard deviations which separates the mean of the LLR assuming the null to be true from the LLR assuming the alternative to be true as

$$\frac{\langle\Lambda\rangle_{s+b} - \langle\Lambda\rangle_b}{\sigma_b} = \frac{s}{\sqrt{b}}, \tag{22}$$

using $\sigma_{s+b} = 4b\ln(1+s/b)^2$ is the RMS of the null LLR. Similarly we can find

$$\frac{\langle\Lambda\rangle_{s+b} - \langle\Lambda\rangle_b}{\sigma_{s+b}} = \frac{s}{\sqrt{s+b}}, \tag{23}$$

using $\sigma_b = 4(s+b)\ln(1+s/b)^2$ is the combined RMS of both the LLR under the null and the LLR under the alternative. By calculating the significances this way we can see directly how they are a measure of the number of standard deviations that observed data is from the expected null LLR.

We show this in Figure 14 (for example LLR distributions) in the upper left plot which corresponds to $s/\sqrt{b}$ and in the upper right plot we show that this is equivalent to the area from the average alternative LLR distribution to infinity, under the null LLR distribution. This

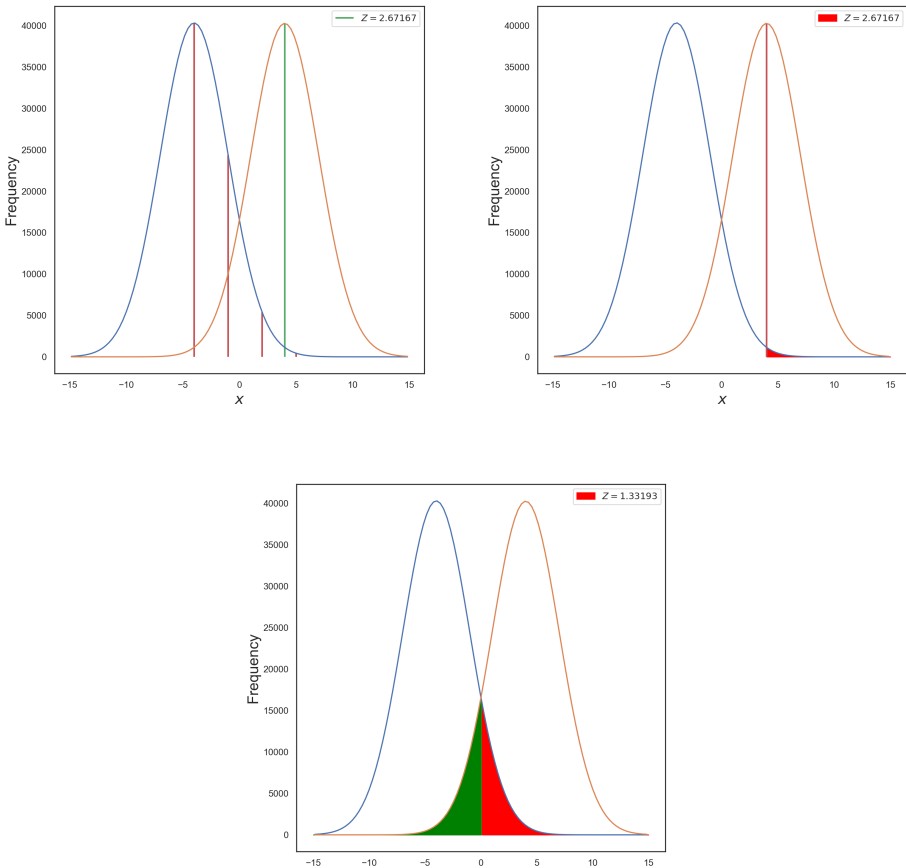

Figure 14: Upper left: visualisation of the naive significance as the number of standard deviations between two distributions. Upper right: visualisation of the significance form the area under the null LLR tail - the agreement between $Z$ in these two plots highlights their equivalence and furthermore their equivalence to asymmetric testing. Bottom: visualisation of symmetric test - notice this $Z$ is half of the above values. All plots made with example toy distributions.

is therefore both the average significance and also the seperation significance obtained under the asymmetric condition. In the lower plot we also show the area under the tail of the null LLR distribution from the midpoint between the two LLR distributions (actually where the area under the two distributions are equal, but for distributions with the same standard deviation this is the midpoint). The average significance calculated this way is equivalent to the symmetric testing condition. If the standard deviations are equal then this significance will be half of the significance that is equivalent to the asymmetric testing condition.

The Asimov significance is easily found by considering a generalised hypothesis test between a background with mean $b$ and some observed dataset with $n$ observed events. If $b$ is known then given $n = \hat{s} + b$ we have the LLR

$$\Lambda = -2\ln\left(\frac{\text{Pois}(b|n)}{\text{Pois}(\hat{s}+b|n)}\right) = 2\left(n\ln\frac{n}{b} + b - n\right). \tag{24}$$

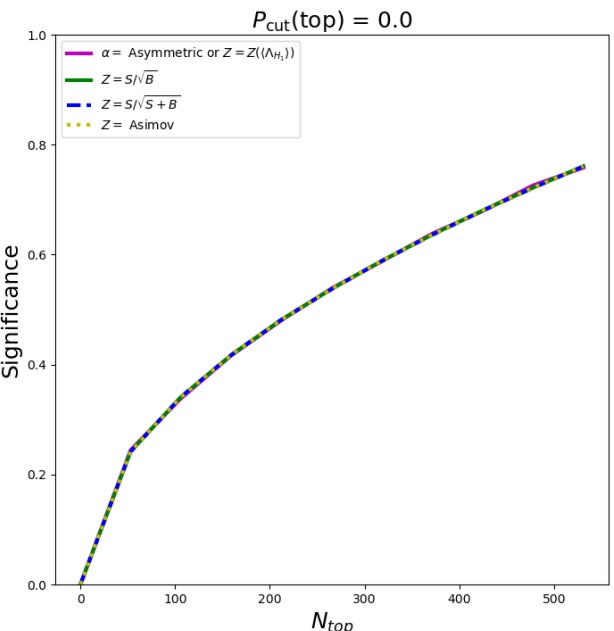

Figure 15: A check (confirmed by matching curves) that the asymmetric separation significance or average discovery significance calculated directly from the LLR with only the Poisson factor are equivalent to the common discovery significances.

It has been shown [34] that under Wilk's theorem

$$
\begin{aligned}
Z &= \sqrt{\Lambda} \\
&= \sqrt{2((\hat{s}+b)\ln(1+\hat{s}/b)-\hat{s})},
\end{aligned}
\tag{25}
$$

yields the Asimov significance. If one were to take the median significance over an ensemble of experiments (i.e. $\hat{s} = s$), then we obtain the median Asimov significance

$$
\text{med}[Z] = \sqrt{2((s+b)\ln(1+s/b)-s)}.
\tag{26}
$$

In the limit $s \ll b$ this reduces to $s/\sqrt{b}$. Note that there an equivalent Asimov significance [79] could be obtained by assuming an uncertainty on the background which reduces to $s/\sqrt{s+\sigma_b^2}$.

As another check of the equivalence between the common discovery significances and the asymmetric separation significance, we compute the $n_\sigma$ from our LLRs from the simple hypothesis task on jet images using only the Poisson factor as shown in Figure 15 using the QCD vs QCD + Top setup. The separation significance $n_\sigma$ obtained under the asymmetric condition is also the same value as the average significance $Z$ obtained from the LLR distributions, which is also the same value as the common discovery significances.

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
