# Peer review of "A simple guide from Machine Learning outputs to statistical criteria"

_SciPost Physics Core, doi:SciPost Phys. Core 5, 050 (2022)_

## Round 1 · Referee Report · Anonymous (Referee 1) · 2022-5-2

Report
My apologizes for the very late report!
This paper describes how to take the output of a neural network and use it for hypothesis testing in the context of collider physics. While the manuscript is mostly well-written, I am not sure what this paper adds to the literature. There is nothing special about neural networks - the paper describes how one takes a fixed observable and then uses simulations to create histograms of the observable to compute p-values. This is common knowledge and so I am unsure what is new in this paper. From the title, I was expecting to see some discussion of how one can use network outputs directly as test statistics instead of indirectly by first making histograms to compute likelihood ratios. This subject is also well-documented in other papers, but there could be room for additional innovation, such as in the area of uncertainty quantification (which is also not really discussed in detail since the entire topic of nuisance parameters are relegated to references). I am sorry if I have misunderstood the paper, but in the present form, I cannot recommend publication in SciPost Physics.
This paper describes how to take the output of a neural network and use it for hypothesis testing in the context of collider physics. While the manuscript is mostly well-written, I am not sure what this paper adds to the literature. There is nothing special about neural networks - the paper describes how one takes a fixed observable and then uses simulations to create histograms of the observable to compute p-values. This is common knowledge and so I am unsure what is new in this paper. From the title, I was expecting to see some discussion of how one can use network outputs directly as test statistics instead of indirectly by first making histograms to compute likelihood ratios. This subject is also well-documented in other papers, but there could be room for additional innovation, such as in the area of uncertainty quantification (which is also not really discussed in detail since the entire topic of nuisance parameters are relegated to references). I am sorry if I have misunderstood the paper, but in the present form, I cannot recommend publication in SciPost Physics.

Author: Veronica Sanz on 2022-06-24 [id 2608]
(in reply to Report 1 on 2022-05-02)Dear referee,
Thanks for reading the paper and providing comments. Note that we have submitted this manuscript to Scipost Physics Core, not Physics. This is a contribution which (we believe) may help readers to navigate between ML outputs and usual statistical outputs, but it is by no means a significant addition to the field. Our purpose is to provide, as the title suggests, a simple guide to perform this translation for supervised and unsupervised ML outputs, and show some of the limitations we found. We hope this clarifies a possible misunderstanding.
Best regards,
V. Sanz

---

## Round 1 · Referee Report · Tilman Plehn (Referee 2) · 2022-6-24

Strengths
The paper offers a very discussion of the way uncertainties could be included in analyses involving neural networks. While I am not sure the topic of the solutions are completely new, I think the paper can be very useful for a phenomenological audience. I especially like the way it discusses the use of anomaly searches, a topic which is not widely discussed in the available literature. In addition, the paper is written for an audience which is different from the usual experimental papers on the topic, so it will be useful for quite a few readers.
Weaknesses
There are not really groundbreaking new ideas in the paper, but there are very nice examples. While for SciPost Physics this lack of new ideas might be a problem, I am fine with publishing the paper in Core. Also, we should keep in mind that lots of model building papers or experimental analyses or ML-studies include no new ideas or in any way unexpected results.
Report
I added whole lot of comments on the presentation in the attached pdf file. They include possible clarifications and also critical comments. The problem is that I cannot upload the file, it is just a Mac-annotated version of the original pdf file, so I am not sure what to do. I will send it to the authors directly for now, to avoid further delays.
Requested changes
Please go through those comments and consider them, in cases where I did not get it, please feel free to ignore them and tell me off...

---

## Editorial Decision

published